# Novel Production Methods of Polyhydroxyalkanoates and Their Innovative Uses in Biomedicine and Industry

**DOI:** 10.3390/molecules27238351

**Published:** 2022-11-30

**Authors:** Guillermo Fernandez-Bunster, Pamela Pavez

**Affiliations:** 1School of Medical Technology, Faculty of Medicine, University of Valparaiso, Campus San Felipe, Camino la Troya S/N & El Convento, Circunvalación Ote, San Felipe 2173310, Chile; 2Centro de Investigación Interdisciplinario en Salud Territorial del Valle de Aconcagua (CIISTE), San Felipe 2173310, Chile; 3School of Biological and Chemical Sciences, Faculty of Medicine and Science, San Sebastian University, Santiago 8420524, Chile

**Keywords:** biotechnology, renewable polymers, biodegradable polymers, polyhydroxyalkanoate, bacterial accumulation, industry

## Abstract

Polyhydroxyalkanoate (PHA), a biodegradable polymer obtained from microorganisms and plants, have been widely used in biomedical applications and devices, such as sutures, cardiac valves, bone scaffold, and drug delivery of compounds with pharmaceutical interests, as well as in food packaging. This review focuses on the use of polyhydroxyalkanoates beyond the most common uses, aiming to inform about the potential uses of the biopolymer as a biosensor, cosmetics, drug delivery, flame retardancy, and electrospinning, among other interesting uses. The novel applications are based on the production and composition of the polymer, which can be modified by genetic engineering, a semi-synthetic approach, by changing feeding carbon sources and/or supplement addition, among others. The future of PHA is promising, and despite its production costs being higher than petroleum-based plastics, tools given by synthetic biology, bioinformatics, and machine learning, among others, have allowed for great production yields, monomer and polymer functionalization, stability, and versatility, a key feature to increase the uses of this interesting family of polymers.

## 1. Introduction

Biomaterials, as definition, are considered as materials that are derived, extracted, or evolved from biological species, with mechanical and chemical properties suitable to be used as material and are biocompatible, namely, the ability of the material to be applied or used over a suitable host, without any harm [1], in terms of not inducing inflammation nor a toxic response when used on a biological system, with an easy degradation and metabolization [2].

Polyhydroxyalkanoates (PHA) are thermoplastic, biocompatible, and biodegradable polymers obtained from several bacterial strains and can be classified as short (3-hydroxybutyrate (PHB) and 3-hydroxyvalerate (PHV)—scl), medium (mcl), and long chain lengths [3]. PHAs have verified applications in industry, agriculture, and biomedical fields. The most common one, PHB, is characterized as a stiff, thermoplastic material with relatively poor impact strength, but adding other scl-chain-lengths may improve the rheological properties of the polymer blend, increasing the number of applications [4]. For example, poly(3-hydroxybutyrate-co-3-hydroxyvalerate; PHBV) has better ductility, strength, and processability, when compared to PHB, while a blend of 3HB and (R)-3-hydroxy-hexanoate (3HHx), P(3HB-co-3HHx) has better flexibility [5].

PHAs produced from microbes are considered an alternative plastic material to avoid a future crisis for the depletion of petroleum-based polymers, but their production costs are higher than petroleum-based polymers [6]. Additionally, and surprisingly, PHA biopolymers may have a rancid odor, depending on how they are produced, explained by the presence of lipid residues and bacterial endotoxins that may be attached to the biopolymer after the extraction, being a handicap for potential applications [7,8].

According to the monomer (or biopolymer blend), PHAs will have different mechanical, rheological, and degradation properties [9]. These biopolymers are mainly produced by fermentation processes, and their structure, molecular mass, and PHA bioaccumulation depend on the carbon source, culture medium, or the PHA synthase [10].

Also, one of the current environmental problems is global warming. Due to this, it is imperative to transition from petroleum-based industries into biorefineries and cell factories to produce compounds of interest, such as chemicals, fuels, and biopolymers from renewable sources. Microplastics are also a current problem caused by the accumulation and overuse of non-degradable petrochemical-based plastics. Therefore, bio-based plastics have been recommended to solve plastic pollution, as well as fossil fuel depletion [11].

## 2. A Little Bit of History

The german hair-care company Wella, in 1992, fabricated the first commercial bottle based on PHBV [12], and from there, PHB and PHBV were used as watertight films of diaper coverings by American companies, while in 2003, several disposable items were manufactured from PHA and commercially distributed in Japan [13]. Biomer, also a German company, developed the production of PHB from *Alcaligenes latus,* manufacturing pens, combs, and even bullets. The Chinese Tsinghua University worked on an *Aeromonas hydrophilahas* strain that produced P(3HB-co-3HHx) PHBHXx, resulting in an appropriate material for binders, nonwovens, elastic wrapping, synthetic paper, thermoformed articles, and medical devices, among others [12]. Similarly, PHA has been used as a material for films, foils, and diaphragms [14]. Other PHAs, such as 4- hydroxybutyrate; 4HB), can be used in medical applications, such as scaffolds for a heart valve, cartilages, tissue, delivery systems, and micro- and nanospheres, among other uses (Figure 1) [15].

Agricultural uses, such as PHA biodegradable nets ((4HB) and polylactic acid (PLA)/PHA-blends), due to their compostability, allow direct disposal of the bioplastic in the soil [16]. An advantage of using PHA is that degradation of the biopolymer does not produce toxic residues then compared to common plastic bags, but also PHA grow bags do not contaminate surrounding water bodies [17], not affect roots, as compared to polyurethane (PE) bags that cause root deformity [18].

The importance of PHA relies on the fact that the degradation products (by a microorganism, such as bacteria and fungi) are not toxic to the environment because the final products are CO_2_ and water under aerobic conditions or as CO_2_ and methane during anaerobic conditions [19,20].

The problem with fossil plastics is the disposal strategy at the end of life: Recycling processes are not available for all petroleum-based polymers and may leak into aquatic and terrestrial environments from microplastics to full-packaging contamination. PHA bottles in marine environments degrade within 1.5–3.5 years, while a petroleum-based bottle may take decades or even centuries to degrade—that disintegration of petro-plastic bottles would take. Additionally, calculations based on life cycle studies have estimated that using 1 kg of PHA instead of common plastics could avoid the production of CO_2_ emissions by 2 kg [21].

Novel processing methods of biodegradable plastics have been developed in recent years, and also the industry is looking for newer, greener, and sustainable products to use in packaging, bottles, and medical applications, aiming to comply with national and international regulations (Figure 2) [22].

PHAs have a great competitor: synthetic fossil-based plastics. They are part of our daily life due to their low cost, easy accessibility, high durability, and flexibility, allowing us to use molding to get any desired shape. Sadly, nonbiodegradable petroleum-based polymer pollution has been one of the biggest environmental problems due to their lack of degradability under aerobic or anaerobic conditions [23].

Production prices, as Koller et al. explain, may be lowered by using cheaper carbon sources on natural PHA producers, by tailoring microorganisms by gene heterologous expression, synthetic biology, systems biology and/or evolutionary engineering, using bacterial strains (such as *Halomonas sp*.) that allows PHA production under the absence of sterility, or by developing novel processes for intracellular PHA recovery [24]. Despite its brittleness, PHB has better UV resistance and mechanical stability, and copolymer blends between PHB and other monomers have improved their biodegradability and compostability [21]. The current biopolymers used by companies are PHB, 4-PHB, PHBV, poly(3-hydroxybutyrate-co-4-hydroxybutyrate) (P(3HB-co-4HB)), and poly(3-hydroxybutyrate-co-3-hydroxyhexanoate) (P(3HB-co-3HHx; Figure 1)), despite the possibility to tailor-made the biopolymer [25].

Different biotechnological strategies have been used for the development of PHA. Taguchi et al. summarize PHA biosynthesis strategies into three generations: The 1st generation is based on fermentation technology on natural PHA producers, manipulating carbon and nitrogen source proportions. The use of recombinant gene technology in non-producer strains was considered the second generation, while evolutionary engineering or “enzyme evolution” was considered the third-generation approach for increasing PHA production [26]. All of these advances were aimed at the same objective: Create novel PHA materials with high yields. The high manufacturing cost had to be reduced to obtain a sustainable industrial process. The first and second generations diminished the production cost but did not compete with the cheap process of petroleum-based plastics [27].

## 3. Novel Uses of Polyhydroxyalkanoates and How to Produce Them

### 3.1. Bacterial Accumulation and Bioremediation

One way to improve the production of biopolymers is by the use of cellular factories, which allow for the production and/or incorporation of monomers in the biopolymer that cannot be added via chemical synthesis. Additionally, cellular factories are less contaminant than using regular chemical synthesis, which requires several steps. As stated previously, monomer incorporation depends on the carbon source used by the producer, metabolic pathways involved in the conversion of carbon source to monomer, and PHA synthases and enzymes related to biopolymer production. In combination, monomers with specific side chains may be chemically modified to increase the functionality and possible applications of biopolymers [28].

To develop novel uses for PHA, understanding the mechanisms involved in PHA production allows researchers to synthesize and develop PHA-based material with desired properties from cell factories, able to produce high amounts of the biopolymer for industrial uses. Biotechnology, including synthetic and systems biology, as well as evolutionary engineering, can help to reduce time, labor, and costs for the production of polyester, aiming to compete with petroleum-based polymers. Historically, heterotrophic bacteria such as *Cupriavidus* spp. and *Pseudomonas* spp. have been used as the main PHA producer [29], while novel accumulators such as cyanobacteria and purple bacteria have been investigated for PHA producers due to their photosynthetic and nitrogen fixation abilities, avoiding the requirements of organic carbon and nitrogen sources [30]. Halophiles have also been used to reduce freshwater consumption and lower contamination risks by the high salt requirements.

Orellana-Saez et al. (2019) described *Pseudomonas* sp. MPC bacterial strain, isolated from Antarctica, as able to grow from sugars, ethylene glycol, and even toxic aromatic compounds, such as toluene, phenol, and chloro-compounds, to accumulate PHA, as well as 5-carbon compounds, such as xylose and arabinose. In bioremediation conditions, the strain may transform phenol to catechol and also degrade benzene, toluene, and styrene, a common material started used in industry. The strain synthesizes a combination of scl-co-mcl copolymers under glucose, while in xylose and arabinose, only mcl is obtained [31].

*P. chlororaphis* subsp. *aurantiaca* produced mcl-PHA from crude glycerol [32] from biodiesel production, having physical-chemical properties that render their high potential for use in the development of wound management materials. The polymer’s low crystallinity and hydrophobicity allow its use for the preparation of flexible and elastic films resistant to water, confirming the potential applicability of these materials for wound dressing applications [33].

One particularity of using PHA accumulators is that they can do two things simultaneously, as demonstrated by Papa et al. (2020), which established a mixed microbial culture as biorefinery to produce biomethane (CH_4_) and PHA simultaneously, aiming to valorize the organic compounds found on municipal solid waste (Organic Fraction of Municipal Solid Waste (OFMSW)) [34,35].

PHAs may be used as a carbon source for post-denitrification and micropollutant co-metabolism. In fact, PHB and PHV are the most common biopolymer used for the removal of phosphorus, chlorinated hydrocarbons, and micropollutants; more specifically, PHB is used as an electron donor source for in situ bioremediation [36]. Santorio et al. (2019) focused on the evaluation of the PHA’s potential use as an endogenous carbon source for denitrification [37], which supports the hypothesis that the lower the PHA concentration within the cells, the lower its use for denitrification, as the biomass prefers to store it as a carbon source [37].

Levett et al. (2019) described that PHA might be accumulated by bacterial strains during enhanced biological phosphorus removal (EBPR), in which, in an anaerobic phase, carbon-based acids within the wastewater, such as volatile fatty acids (VFA), are assimilated by the microorganisms and stored as PHA by phosphate accumulating organisms. In the following step, under oxygen conditions but carbon starvation, the biopolymer is used as a carbon and energy source for cell growth and to uptake and store phosphates and glycogen. The research group additionally tested drug removal, such as benzotriazole, 5-methyl-1H-benzotriazole, carbamazepine, ketoprofen and diclofenac, and ibuprofen, among others. For example, under different conditions, such as low phosphorus concentrations, ibuprofen was removed faster than higher phosphorus concentrations, but that degradation would depend on the bacterial composition [38]. Also, worth to mention that bacterial strains in biofilms may be grown on plastic carriers that are suspended in the system, which require biodegradation [39].

Amino acids and vitamins on the diet of the PHA producer *Alcaligenes* sp. NCIM 5085, using different carbon sources, such as sugar cane molasses, fructose, and wheat bran, affects the properties of the PHB: a different PHB was obtained, characterized by high molecular weight and crystallinity, with higher thermal stability than regular PHB. Sugar cane molasses with methionine supplementation increased the PHB maximum yield, according to the authors, because amino acid biosynthesis requires more energy (ATP) than other amino acids [40].

Natural PHA-producing organisms accumulate PHA as granules surrounded by lipids and proteins [28,41], as well as non-native recombinant producers. But, heterologous hosts do not necessarily use the same metabolic regulation as natural producers, making them possible better for large-scale production.

PHA production pathways, such as the β-oxidation pathway, may be diverted into PHA production via modifications on enzymes, such as enoyl-CoA hydratases (PhaJ) [42], FadB homologs (YfcX, PaaG, PaaF, and YdbU), among others [28]. A known example is the work of Taguchi et al., who produced poly(lactide-co-3- hydroxybutyrate) from non-related carbon sources. Specifically, the group added a synthetic metabolic pathway to transform pyruvate to lactate and then to lactyl-CoA, a substrate able to be used by engineered PHA synthases to copolymerize with 3HB monomers [43]. This research was groundbreaking in using engineered PHA enzymes with broad substrate specificity. In terms of evolutionary engineering, research is usually based on the main PHA synthase or PhaC enzymes, which catalyze the polymerization of hydroxyacyl monomers, such as PHB, PHV, or others, to produce PHA polymers. There are different PhaC enzymes, depending on their substrate. For example, PhaC from *Ralstonia eutropha* has specificity to scl-PHA monomers, while PHA synthases from *Pseudomonas* sp. have substrate specificities to mcl-PHA monomers [44]. Laboratory evolutionary techniques may be used to generate novel PHA synthases with increased affinity and/or substrate specificity [45].

Natural PHA synthases only may use 3-, 4-, 5-, and 6-hydroxyacyl-CoA groups as substrates. By evolutionary engineering, several studies were attempted successfully to use lactate as a monomer in a 3HB copolymer by evolving propionyl-CoA transferase (Pct) and the phaC [11,46].

To engineer a microbial host to obtain PHAs using cheap carbon sources crucial, such as sugarcane molasses and sucrose, among others, can be done by metabolic engineering on the natural hosts by adding the PHA-producing genes (Table 1), such as acetyl-CoA acetyltransferase (PhaA), acetoacetyl-CoA reductase (PhaB), and PHA synthase (PhaC) using acetyl-CoA as the starting metabolite on a heterologous host, by evolutionary engineering, among others (Jae Park, 2015). Sohn et al. developed a recombinant *E. coli* that use sucrose as a carbon source that produces 3HB and P(3HB-co-Lactic Acid) by the expression of the *sacC* gene, encoding for secreted β-fructofuranosidase that mediates sucrose hydrolysis into glucose and fructose [11].

Genetic engineering and directed evolution have based their efforts on class I and class II PHA synthases due to their being formed as a single subunit, while those of class III and class IV are multisubunit synthases. While some PHA synthases are well characterized, allowing for pinpoint modifications, a great number of PHA synthases do not possess a structural model for directed mutagenesis to improve activity, substrate specificity, and stability of these enzymes, requiring more “irrational approaches” [26].

Other molecular techniques to improve PHA synthesis are given by synthetic biology, fiving tools such as ribosome-binding site (RBS) optimization, promoter/repressor engineering, chromosomal integration, cell morphology engineering [47], and cell growth behavior re-programming, among others. The gene-editing tool CRISPR/Cas9 has been demonstrated to be an excellent tool for regulating PHA metabolic fluxes and PHA pathway optimization. Synthetic biology methods and genome-editing tools are the present and future of PHA production, which will allow us to obtain tailor-made biopolymers, decreasing production costs. The up-scaling processes to obtain higher amounts of PHA have been difficult to overcome in the past. Molecular approaches, such as RBS optimization, promoter/repressor optimization, gene deletion, or expression-weakening methods, have been developed in non-model organisms [48]. For instance, different PHA may be obtained by weakening the beta-oxidation cycle in *Pseudomonas putida* and *Pseudomonas entomophila* [49]. Other strategies include the downregulation of branch products, controlling oxygen availability, regulation of NADH/NAD+ ratios, increasing oxygen availability, and regulation of the PHA molecular weight and granules, forming larger PHA granules and regulating PHA molecular weight also help to enhance PHA production. Further details may be found in a remarkable review of synthetic biology tools for PHA production [48].

Enzyme evolution applied to PHA production has been considered effective in altering enzymatic function and/or carbon source flexibility. Despite natural evolution having allowed us to find novel bacterial strains able to accumulate PHA for survival as carbon source storage, enzymes are not necessarily adequate for industrial uses. Changes in carbon source, stress resistance, pH, and temperature, among others, may be currently modified and tailored for the required reactor. Because of this, the concept of “evolutionary engineering” appeared, currently named “directed evolution” [50], by gradually changing carbon source concentrations and growth conditions, aiming to adapt the bacterial strain to “harsh” conditions. This approach may be combined with site-specific mutagenesis but requires structural information to select an adequate site for the mutation. Taguchi et al. (2004) explain that also an “irrational” directed evolution approach can be used by random mutagenesis with selection and/or screening, which may generate the desired change or other non-planned features, generating novel biological functions by molecular evolutionary mechanisms [26].

Random mutagenesis is considered a directed evolution strategy that aims to introduce random (hence the name) point mutations into the genome. The type of mutagenesis may be divided into five categories: transitions (substitution of a purine nucleotide for another or a pyrimidine for another one), transversions (substitution of a purine for a pyrimidine or vice versa), deletions (removal of one or more nucleotides), insertions (addition of one or more nucleotides) and inversions) and inversions (a full 180° rotation of a DNA segment) [51]. These types of mutations may be produced by treating the bacteria or its nucleic acids with chemical mutagens [52], by error-prone PCRs [53] and saturation mutagenesis, among others.

Evolution systems can be applied to all the enzymes involved in polyhydroxyalkanoate biosynthesis. PhaC, the main synthase, for example, catalyzes the polymerization of 3-hydroxy-acid-CoAs, into PHA granules. The members of this protein family are categorized into three major classes in terms of the primary structures deduced from their DNA sequences and the substrate specificity of the enzymes [26]. In general terms, the effectiveness of PhaC is related to the characteristics of the generated biopolymer, such as molecular weight, monomer composition, and even the PHA cell content.

The concept of in vitro evolution of the PHA synthase starts with an “original” PHA synthase (or prototype). After random mutagenesis, a pool of mutants that goes under a high-throughput screening, such as fluorescence-activated cell sorting (FACS), looking for evolvants with increased activity, thermostability, and change of substrate specificity, among others, which may be used as an efficient production system of PHAs with desirable properties. Taguchi et al. used an error-prone DNA polymerase for mutagenesis during PCR and developed two assay methods to evaluate polymer accumulation in a recombinant *E. coli* strain, focusing on the mutant generation of the phaC synthase gene [26]. Using another approach, Rehm et al. used a single gene shuffling to the coding region of phaC to generate a completely shuffled DNA fragment with mutations [54]. Liu-Tzea et al. analyzed the catalytic domain of the PHB polymerase in *Ralstonia pickettii* T1, concluding that the Asn at position 285 (N285) participates in the cleavage of ester bonds in a study performed using evolutionary engineering, controlling its activity [55]. Other studies about evolutionary engineering on *Ralstonia pickettii* T1 can be found in several papers [56,57,58].

In *Aeromonas caviae*, a single aminoacid substitution on the PhaC synthase went under study, which is able to produce the polyhydroxyalkanoate copolymer [P(3HB-co-3HHx)], demonstrating that a change from asparagine 149 by serine or aspartate 171 by glycine (D171G) presented higher enzymatic activity. Combining evolutionary engineering and gen insertion in *Ralstonia eutropha*, the synergistic effect of both aminoacid changes allowed the use of longer 3HA units from octanoate and soybean oil as carbon sources. When fructose was used, the PHB synthesized had a higher molecular weight [59].

The structural function of PhaC will help in understanding accumulation and biopolymer composition, giving light to future protein engineering works to improve PHA yield and biopolymer [60]. Another way to cheapen the costs of PHA production is by using mixed microbial cultures (MMCs), due to the lack of sterilization required and simplifying the process control, as well as allowing the use of low-cost substrates, such as agro-forestry residues, industrial by-products and wastes [61].

The second-generation strategy does not rely solely on the addition of the PHA genes but also the selection of them: Carbon sources and substrate specificity (given by the phaC synthase gene) affect the chemical structure of the comonomer units, monomeric composition, and molecular weight. Between abnormal climate change, global warming, plastic waste accumulation, and other anthropogenic environmental issues, several metabolic engineering strategies may be used to develop efficient PHA-producers microbial host strains.

Foong et al., by synthetic biology, enhanced the PHA productivity of the marine photosynthetic purple nonsulfur bacterium *Rhodovulum sulfidophilum*, a weak halophile previously used as a platform for the production of biohydrogen, PHAs, and spider silk [23]. The problem with the bacterial strain was the low substrate affinity of PHA synthase, so by genome-wide mutagenesis induced by ethyl methanesulfonate together with ultraviolet irradiation dosages and by a single-cell mutant screening of high PHA-accumulating mutant *R. sulfidophilum*, they obtained a PHA producer with higher PHA yields [6]. They chose genome-wide mutagenesis considering the highly complex cellular metabolic processes within a versatile photosynthetic bacterium that are affected by light, oxygen, and carbon source availability [62,63]. The research demonstrated that a synthetic biology-based strain improvement approach, using genome-wide mutagenesis and high-throughput FACS for selection, simplified the mutant generation and screening processes for complex microorganisms, enhancing up to 1.7-fold higher volumetric PHA productivity and a faster accumulation speed.

Other studies related to the biosynthesis of PHBHHx have been done in engineered *C. necator* strains using CO_2_ as the sole carbon source. Different enzyme, carbon and nitrogen source combinations have shown that PHA composition may be regulated, highlighting that by using recombinant (R)-enoyl-CoA hydratase (PhaJ) with different substrate specificities [64].

A surprising heterologous host is *Halomonas* sp. TD01, a fully-genome sequenced halophile bacteria isolated from China, can be grown under high salt concentrations and pH, allowing an open fermentation without contamination [65]. Unfortunately, the bacterial strain did not possess a controllable repression system for gene expression has slowed down more applications for *Halomonas* sp. TD01. Specifically, propionic acid turns into propionyl-CoA, which can enter the PHBV synthesis pathway (from foreign bacteria) to form 3-hydroxyvalerate monomers. Gene editing tools, such as CRISPRi (clustered regularly interspaced short palindromic repeats interference) derived from the CRISPR/Cas9 system, provide an efficient method for targeted gene repression [66]. Basically, in the CRISPR system, the Cas9 protein (RNA-guided DNA endonuclease) binds to a small guide RNA (sgRNA), forming a protein-RNA complex that will bind to the targeted sequence [67]. Mutations in the Cas9 protein can produce a Cas9 protein with DNA-binding properties. This feature may be used to target specific DNA, block transcription, and interfere with RNA polymerases, among others [68], a process named CRISPRi. The molecular technique was used to regulate PHA production by repressing multiple genes or multiple targets on one gene [69]. Wei Tao et al. developed a CRISPRi system for the non-model organism *Halomonas* sp. TD01 was successfully constructed and enhanced PHB expression by the repression of the ftsZ, prpC, or gltA genes. ftsZ encodes for a bacterial fission ring formation protein for elongated cell morphology and longer cell sizes, prpC gene encodes for a 2-methylcitrate synthase that regulates the 3HV monomer ratio in PHBV copolymers. Finally, the repression of the gltA gene, which encodes for citrate synthase, shunted a greater proportion of acetyl-CoA from the tricarboxylic acid (TCA) cycle to PHB synthesis, being obtained 8% more biopolymer by the repression of the gltA gene [47]. Other gene editing tools, such as CRISPR-Cas12a, allow for the simultaneous insertion of genes into multiple specific loci of the heterologous host genome, helping to edit microbial genomes faster [70]. At the pace of current progress, it is feasible to think of more novel and efficient synthetic biology techniques that will be developed to edit, design, construct or regulate the PHA synthetic pathways in natural and non-natural production strains.

Another heterologous host with similar properties as *Halomonas* sp. TD01 is the *H. bluephagenesis* TD01 bacterial strain, which can grow rapidly, as well as under high salt concentrations and alkaline pH, making it suitable for cultivation during nonsterile, open, and fermentation processes [71]. By using the bacterial strain, the group deleted the chromosomal phaC gene by using CRISPR-Cas9 while also using a plasmid expressing the phaCJ gene cloned from *Aeromonas hydrophila*, obtaining PHB-HHx.

### 3.2. Packaging: More Than Food

Most PHA research is based on two main topics: Biomedical applications and food packaging, but biomaterial may also be used for feeding purposes. Feeding aquatic organisms with PHB was not deadly to animals [72], even positively affecting the growth rate, survival, and disease resistance of aquatic animals. PHB works as an additive as a supplier of short-chain length fatty acids to maintain an adequate microbiome in the gastrointestinal tract [73]. As in bacteria, PHB may be used as a carbon (energy) source by European sea bass [72], rainbow trout, and Chinese mitten crab [74], among others. Also, polyhydroxybutyrate is used as an antibiotic alternative against *Vibrio campbelli* in shrimps by stimulating the expression of heat-shock proteins 70 [75].

Wang et al. (2019) used PHB (1% to 4%) as diet supplementation, producing a feeding pellet to be used in large marine yellow croakers and sensitive weaned piglets [76]. Also, it appears to be species-specific because no difference was found in the growth of Nile tilapia [73,77].

Also, combinations with other biopolymers (named composites) may be obtained. PHB/cellulose composites have less brittleness and are stable, which may be used for food packaging [1]. Other examples are blends are polylactic acid (PLA), cellulose, and nanofibers, which can use PHB as a carrier system to develop compatibility between PLA and the nanofibers, improving the properties of the material, being used in packaging and construction material [78]. Other combinations, such as cellulose nanocrystals and PHB as bionanocomposites, improved UV barrier properties and reduction in water vapor permeation, with potential applications in the packaging [79], and the development of a PHB/cellulose cardboard with good mechanical properties that could be used in food packaging and the agriculture industry [80].

A nontoxic, stretchable, biocompatible and biodegradable polyurethane was prepared, characterized, and evaluated for biomedical applications based on L-lysine diisocyanate with poly(ethylene glycol) and polyhydroxyalkanoates (LPH) of different molar ratios were synthesized, demonstrating the viability of the LPH scaffolds for many biomedical applications [81].

Goonoo et al. (2017) developed novel blend films of anionic sulfated polysaccharides kappa-carrageenan (KCG) and fucoidan (FUC) derived from seaweeds with semi-crystalline PHB and PHBV and tested them in cells. Summarizing, mixtures presented higher surface hydrophilicity, higher water uptake, and immiscibility of polymer components, giving rise to an improved biological response of PHBV/KCG blends toward fibroblast growth [82].

The addition of silver nanoparticles (AgNPs) to generate materials with antimicrobial properties is promising for the food storage field [83], offering a strong antimicrobial activity against the food-borne pathogens *Salmonella enterica* and *Listeria monocytogenes*, which makes them potentially suitable for active coatings and packaging applications. Interestingly, the presence of the AgNPs did not impair the profile of biodegradation of the microbial polymer [84,85].

Requena et al. (2019) tested the antibacterial effect of PHBV films with oregano or clove essential oil, or their main compounds, carvacrol, and eugenol, respectively. The analysis was carried out in food matrices and in vitro tests for *Escherichia coli* and *Listeria innocua* [86].

ZnO nanoparticles can be used to enhance the properties of a PHB matrix, increasing the thermal stability and degree of crystallinity of the PHB matrix [87,88], as well as presenting antibacterial (Gram-positive and negative) properties, as well as a decrease in oxygen permeability, water vapor, and water uptake of the PHB host matrix, which may be used as antimicrobial plastic packaging [89]. Adding ZnO nanoparticles in a copolymer of PHBHHx allows for an increase in UV absorption, melting temperature, and thermal stability, making it suitable as a UV-blocking material or as packaging [90].

A blend of PHB/poly(butylene adipate-co-terephthalate; PBAT), together with the antimicrobial compounds of 1-allylhydantoin (AH) and perfluorooctyl acrylate (PFA), gave, as a result, a PHA-based membrane with ductility, stability, and hydrophobicity adequate for packaging applications [91]. Other mixtures, such as PLA/PHB [92], polyhydroxyalkanoates/cheese whey/gelatin (PHAs/CW/Gelatin) [93] and PHAs/nanokeratin [94] and PLA-based thermoplastic polyurethane (PLAPU) [95] also possess great features to be used as packaging material [86].

Active compounds may be volatile and may also affect the organoleptic properties of the food; for example, oils immobilized in silica changed the aroma of juice samples. For example, Figueroa-Lopez et al. (2020) used different concentrations of eugenol (C_10_H_12_O_2_)—commonly known as clove oil, recognized by its antibacterial properties against Gram-negative and positive strains [84]—and incorporated it into ultrathin fibers of PHBV to create antimicrobial monolayers, testing it against *Staphylococcus aureus* and *Escherichia coli*, with positive results [85]. Grape seeds lignin (1–5 wt%) positively affected the PHB/PHA films by improving their gas barrier properties, thermal stability, antioxidant activity, and biodegradability. As an end-product, the biodegraded polymer of PHA/PHB/lignin did not show toxicity [96]. Xavier et al. (2015) studied the incorporation of vanillin in PHB films for use against fungal strains, such as *Aspergillus fumigatus*, *A. niger*, *A. ochraceus*, and *Penicillium viridicatum*, and also against common pathogenic strains, like *Escherichia coli*, *Salmonella typhimurium* and *Staphylococcus aureus* [97]. Eugenol may also be added for its antifungal efficiency in the treatment of infections and/or food packaging [98].

### 3.3. Drug Delivery of Natural Compounds

The biopolymer storage technology, whose hydrophobicity may be configured, allows researchers to encapsulate organic compounds and tailor the biocapsule according to the compound’s properties. By having an engineered strain that produces a drug and encapsulation methods, for example, lycopene and PHB, respectively, encapsulation will be produced in vivo, increasing efficiency and lowering production costs. Additionally, other drugs may also be considered if their biosynthetic pathways are identified to be added to the *E. coli* strain. This type of methodology, commonly named “one-pot,” saves time and may avoid the use of organic solvents to extract the biosynthetic compound [99].

Heterologous gene expression in suitable hosts may be used for compound encapsulation. Fusing enzymes with PhaC allows the production of PHA granules with active enzymes on the surface, yielding various applications in scouring, bioseparation, imaging, and biomedical applications [100]. Former immobilization methods, such as cross-linking and encapsulation, require several steps that could negatively affect enzyme activity [101]. Tan et al. (2019) combined this methodology to add the genes for a tyrosinase (from *Verrucomicrobium spinosum*—TyrV) and immobilized it on the surface of PHA granules (also by the addition of the phaC synthase gene), where the resulting PHA-TyrVs nano-granules demonstrated L-DOPA (3,4-dihydroxyphenyl-L-alanine), a promising drug for Parkinson’s disease [102], forming monophenolase activity, and improving L-DOPA production [100,101,103,104]. The approach has been used before for the immobilization of polygalacturonate lyase (PGL), beta-galactosidase (LacZ) [54], N-acetylneuraminic acid aldolase (NanA) [105], immunoglobulin G (IgG)-binding domain (ZZ) [101], and mycobacterial antigens [106], among others.

Wong et al. (2020) developed a novel technology to use bioengineered PHA particles on engineered bacteria for protein immobilization by adding functional proteins by the SpyTag/SpyCatcher technology of protein ligation. This methodology relies on the production of PHA spheres with a specific domain described as the SpyCatcher domain and synthesizing a SpyTagged target protein that may be ligated to the PHA itself [107]. This strategy, according to the authors, is proposed as a versatile toolbox for PHA sphere functionalization with biomedical and industrial applications [108].

Levett et al. (2019) used it to deliver dicyandiamide (DCD) in PHBV pellets, a heat-sensitive stabilizer for nitrogen fertilizers, aiming to use it in tropical agriculture, characterized by high temperatures and fast microbial degradation [38]. Nitrogen fertilizers are described as very inefficient, and more than half of the fertilizer is lost in the environment and in terms of ammonia and urea volatilization and microbial transformations [109], so nitrification inhibitors, such as DCD, may help to decrease nitrogen losses and, simultaneously, increase plants’ nitrogen uptake [38]. The research group stated that PHAs suffer enzyme-catalyzed erosion at their surfaces as the degradation process in soil [110]. The main idea to use PHA to encapsulate hydrophobic molecules (i.e., compounds that repel water) is explained because biodegradable biopolymers such as PHA allow a controlled release of the encapsulated compound (such as the natural compounds carotene, morin, and the anti-cancer compound curcumin) by monitoring the degradation of the polymer [100], such as beta-carotene in mcl-PHA [111].

Latos-Brozio & Masek (2020) studied the degradation process of PHA biocomposites with natural antibiotic compounds. This is a very interesting idea, but PHA degradation depends on bacterial strains to degrade the polymer, so how biodegradation works when an antibiotic is added? Not only do microorganisms participate in the degradation, but also elevated temperatures and high humidity are adequate conditions for the two main mechanisms of degradation of polymeric materials: Hydrolytic degradation (a chemical-based process that breaks bonds within the polymer) and the biological-based enzymatic degradation by microorganisms [112].

Research groups used simultaneously the PHA bioaccumulation and drug encapsulation properties in the biopolymer-producing bacterial strains [113]: Liu et al. (2020) proposed and reported a methodology that allows the in vivo storage of hydrophobic products in/on biopolymer bodies in *E. coli*, using PHB to encapsulate lycopene, a red carotenoid hydrocarbon that can be found in red fruits and vegetables. After in vitro tests and in vivo characterization, they found that lycopene formed aggregation bodies in bacteria that did not produce PHB. The co-production of both lycopene and PHA titers simultaneously was positively correlated; for example, if a cell had high PHB content, it tended to have also a high lycopene content [99].

Modjinou et al. (2020) combined PHBHV with the monoterpene linalool extracted from spice plants, aiming to develop an antibacterial membrane, increasing the mechanical resistance flexibility and elongation at break. It is worth mentioning that linalool addition was by using a tetrafunctional cross-linking agent, and also the PHA-Linalool material presented anti-adherence properties against *Escherichia coli* and *Staphylococcus aureus* [114], while Latos-Brozio & Masek (2020) et al. studied PHA and PLA matrixes with the plant functional additives, (+)-catechin and polydatin, an anticancer/inflammatory/oxidant agent [115] to improve the biopolymeric material properties, obtaining a polyester with higher resistance to oxidation and degradation under the influence of UV radiation [112,116].

Improvement of PHAs polymer properties by adding functional groups could be a good approach to increase their biodegradability, economic value, and important applications in the medical field, as shown by Bhatia et al. (2019), who added ascorbic acid as an antioxidant to PHBV and obtained better mechanical properties and biodegradability [7].

PHA beads are being tested for industrial purposes, such as degrading dyes generated in the textile industry [117] or to support enzymes that synthesize molecules for food purposes, such as D-allulose, a potential sweetener [118]. Ran et al. (2017) reported that PHA bound to the enzyme PGL (Alkaline polygalacturonate lyase), one of the pectinolytic enzymes used for the bioscouring of cotton fibers, biodegumming, and biopulp production, resulting in a promising approach to immobilization of PGL in vivo, contributing to the wider commercialization of this environmentally friendly biocatalyst [105].

PHAs have been used to encapsulate bacteriophages [119], but no description can be found of bacterial immobilization. Gonzalez et al. (2020) went beyond this and encapsulated microorganisms within polyhydroxyalkanoate (PHA)-based microcapsules (MPs). At first, the group obtained a spherical PHA from a *P. putida* KT2440 strain by using a modified double emulsion solvent evaporation technique based on water/oil/water phase separation, testing a microencapsulation method on PHA by encapsulating the *Bdellovibrio bacteriovorus* bacterial strain [120].

Interestingly, double emulsion technology using mcl PHA may be used for microbial encapsulation with a controlled release while protecting the bacterial strain from environmental stress. This technology does not rely on a certain bacterial strain; instead, bacteria of biotechnological interest may be encapsulated for their use under stressful conditions or as antibiotics.

### 3.4. As a Biomaterial

Why use PHA? Companies are looking for materials that may replace metal parts with polymer-based materials. Polyhydroxyalkanoates may be prepared to have different morphologies and thermal/rheological properties. In terms of biodegradability, the material may be contaminated with other compounds, such as paraffin, in which, under the influence of an environment rich in enzymes, the compound may accelerate biodegradation, but under natural conditions, the contaminant slowed the biodegradation. Also, 3D printing increased the applications of biopolymers by allowing the preparation of specific pieces using the material without requiring molds. Additionally, they stated that the printing orientation of individual parts of the container might influence the properties of the material, individually for each region, in which every individual region had its degradation rate. One of the advantages of 3D printing is that researchers can create any complex shape.

Current technologies have been used to increase the applications of PHA: Gonzalez Aseujo et al. (2018) tested a polylactide/PHA material by using 3D printing and characterized its properties, including behavior during waste disposal [121].

Pop et al. (2020) used a PLA/PHA/Bamboo Fill filament for 3D printing and tested it for tensile, compression, and three-point bending strengths. The more interesting results related to PHA filament were that in pieces subjected to compression, the PLA/PHA/BambooFill materials were recommended in solid configuration, while the cylindrical architecture conferred better compressive strength compared to the standard architecture. In terms of impact resistance, the tensile strength of a 3D-printed sample will depend on the mass of the specimen for all materials [122].

The most recent research has focused on PHA and COVID. Why? Historically, the use of non-biodegradable plastics has increased from 245 million tonnes in 2008 to 368 million tonnes in 2019 [123], even estimated that nearly 1.6 million tonnes per day of plastic waste were produced due to the SARS-CoV-2 pandemic outbreak: Personal protective equipment (PPE), such as surgical facemasks and shield, latex or nitrile gloves, goggles, shoe covers, gowns, among others, are also considered as plastic waste [124].

One of the problems was the pollution produced by single-use face masks. By using 3D printing, Zgodavová et al. (2021) tested different biopolymers to test their biodegradability, being PHA selected by its environmental sustainability [125]. As PHA material, they used PHA BioWOOD Rosa 3D, a 100% natural biopolymer that decomposes without oxygen and water, with a wooden smell, unlikely other PHA biopolymers. This research is recommended by the reviewers because it shows the whole process required to tune and set up 3D printers for PHA uses. The group focused on the face shield frames. By following the protocol, the groups proposed that this roadmap can be applied to new product development, even in biotechnology, for a wide range of 3D technologies.

Other bio-composites based on polyhydroxyalkanoates (PHAs) and fibers of the aquatic plant *Posidonia oceanica* (PO) have been developed that can be suitable to manufacture items usable in marine environments, for example, in natural engineering interventions and represent an interesting valorization of the PO fibrous wastes accumulated in large amounts on coastal beaches [126].

Also, PHA-nanoclay blends were developed to improve the biopolymer properties, and developed nine nanobiocomposite materials, based on a linear PHB, poly(3-hydroxybutyrate)-co-poly(4-hydroxybutyrate; P3HB-co-P4HB); and nanoclays, aiming to customize copolymers to obtain semicrystalline copolymer structures designed to have tailored melting points, as well as controlling its brittleness, obtaining a PHA-nanoclay blend with better mechanical properties, also eliminating the characteristic odor of PHA [8].

Yu et al. (2019) proposed the synthesis of a novel fluorescent material by using highly efficient rare-earth, using PHA as a scaffold [127], as an alternative of Quantum Dots, described as the new-generation in fluorescent materials, which has the unwanted feature of releasing free ions and low solubility [128]. They used an engineered *Halomonas bluephagenesis* for PHA accumulation, and then N-acetyl-L-cysteine (NAL) was added by UV radiation and then combined with rare-earth material. As a result, they got a functionalized NAL-PHA material combined with the rare earth fluorescent material, with intense photoluminescence under UV excitation [71,129,130].

Most PHA research is focused on obtaining biopolymers from natural sources. Kageyama et al. (2021) went beyond and designed their biopolymer, namely poly(2-hydroxybutyrate-b-3-hydroxybutyrate [P(2HB-b-3HB)]), by using an engineered *E. coli* strain expressing a fused (chimeric) PHA synthase PhaCAR, as part of the novel strategy to use artificial PHA to improve the characteristics of the biopolymer by using unusual monomers that show distinct and unique properties [131]. The PHA synthase PhaC1PsSTQK [132] was discovered several years ago and incorporates different 2HA units, such as lactate and glycolate. Despite it being possible to obtain “artificial” PHA blocks, their biodegradability and cytotoxicity are under investigation but open an era in synthetic biology by being able to develop one’s “own” PHA synthase.

### 3.5. Applications of Electrospinning Technique on PHA

Electrospinning has garnered attention in recent years due to its potential for applications in various fields. Electrospun fibers have shown great applicability for novel materials in tissue engineering, wound healing, and bioactive molecules delivery, as well as sensor, filtration, composite reinforcement, and nanoelectronic applications. Electrospinning consists of a syringe through which a polymer solution is pumped, a high-voltage source, and a collector. The pendant drop of polymer solution held by surface tension forces at the tip of the syringe is electrified upon application of high voltage. As a result, electrostatic repulsion is established between like charges within the polymer solution, resulting in Coulomb forces due to the external field. When the strength of the electric field exceeds a threshold value that overcomes the surface tension of the polymer solution, a jet is produced from the pendant drop. The jet undergoes stretching and whipping while traveling toward the collector. The solvent evaporates during this process, and then a solid non-woven fibrous matrix is deposited on the collector. Several diverse polymeric materials have been electrospun, including synthetic and natural polymers. Among natural polymers, polyhydroxyalkanoates (PHAs) have useful properties such as biodegradability, thermoplasticity, biocompatibility, and non-toxicity [133].

Optimally post-processed electrospun fibers exhibited similar rigidity to conventional compression-molded PHA films but with enhanced elongation at break and toughness. Films made by electrospinning technique have, for instance, application interest in the design of barrier layers, adhesive interlayers, and coatings for fiber- and plastic-based food packaging materials [134], for example, the development and characterization of oxygen-scavenging films made of PHB containing palladium nanoparticles (PdNPs) prepared by electrospinning followed by annealing treatment at 160 °C. using hexadecyltrimethylammonium bromide (CTAB) as the dispersing aid. As a result, the PHB/PdNP nanocomposites containing CTAB offer significant potential as new active coating or interlayer systems for application in the design of novel active food packaging structures [135], while graphene-decorated silver nanoparticles (GAg) were incorporated into the fibers of poly-3 hydroxybutyrate-co-12 mol.% hydroxyhexanoate (P3HB-co-12 mol.% HHx), demonstrated significant reduction of *S. aureus* and *E. coli* as compared to solely to PHA. Therefore, the as-spun PHA/GAg nanocomposite may feasibly be efficient in the treatment of chronic wounds and sanitizing applications [136].

Future studies will focus on the application of PHB microfibers in several technological fields; for example, stearic acid-modified bilayers produced from the micro-nano-fibrous of SiO_2_ and PHBV composites showed very low water droplet sticking, which was possible to obtain superhydrophobic micro-nanofibers from PHBV-SiO_2_ [137].

### 3.6. Cosmetics

In cosmetic products, microplastics derived from toothpaste and cleansers, packaging, or face masks, have increased the environmental problem threatening whole ecosystems and human health, exacerbating the plastic problem to the extent that it is now one of the most serious global crises [138]. Additionally, not all petroleum-based plastics can be recycled, which solely only six types of plastics (polyethylene terephthalate, high-density polyethylene, polyvinyl chloride, low-density polyethylene, polypropylene, and polystyrene) may be partially recycled [139].

In biotechnology terms, the concept of accelerated aging is helpful in determining the lifespan or shelf-life of the product at an accelerated pace. Usually, research groups tend to study the rheological and mechanical properties of the biopolymer blends, but storage and shelf-time should also be considered for the market application [140]. In cosmetic products, also it must be considered physical and chemical stability under the country’s standards of safety in terms of storage [141] and/or transportation [140]. The challenge is to find adequate biopolymer cosmetics packaging that degrades even with residues of their content [141].

Compatibility tests must be considered before going to market to determine the interaction of the cosmetic formulation with the packaging in natural and shelf conditions. The ideal packaging should not react nor affect the cosmetic formulation and vice-versa, but also, physical conditions, such as appearance, color, and odor, should be taken into account. Also, depending on the material (PLA vs. PHA/PLA), paraffin (as a cosmetic compound) may accelerate or slow down degradation, among other statements. In general terms, the PLA/PHA blend demonstrated better compatibility with the cosmetic formulations [141].

PHA with starch blends has been used to prepare flexible films in beauty masks that can release starch (or other molecules) after wetting [142]. This strengthens the effort towards caring about raw materials and waste, positioning biotechnology as a promising science to impact the production of beauty ingredients and/or final products [143]. Renewable raw materials may protect the skin from contaminants, as well as modulate the skin’s microbiota, namely the ecological community of microorganisms that colonize our body, adding to the ease of biodegradability and skin biocompatibility [144]. Also, it can be commercialized in a dry state without requiring preservatives [145]. For example, the structural, mechanical, antioxidant, and cytocompatibility properties of membranes prepared from the polyhydroxyalkanoate and arrowroot (*Maranta arundinacea*) starch powder blend demonstrate the enhanced functionality of arrowroot starch/polyester-based membranes for applications in the fields of drugs, food, cosmetics, and biomedical engineering packaging material [142,146].

### 3.7. As Sensor

PHB films may be used as a membrane to incorporate hemoglobin to boost the electron transfer rate of this protein, and other modifications, such as adding peroxidases and/or using pyrolytic graphite electrodes [20,147]. Phukon et al. (2014) used a hybrid nanocomposite combining 3-hydroxyvalerate (3HV), 5-hydroxydecenoate (5HDE), and 3-hydroxyoctadecenoate (3HODE) to be used as a biosensor was used for the quantitative detection of artemisinin in body fluids, combining gold nanoparticles and horse radish peroxidase [148].

Stojanovic et al. (2020) used polyhydroxyoctanoate (PHO) from a *Pseudomonas putida* KT2440 strain, prepared artificial saliva and simulated gastric fluid, aiming to evaluate it as a biosensor. In this particular example, they used pH as a reference because both liquids tested differ significantly in their pH values (Saliva 6.2–7.6; gastric fluid: 1–2). When the polymer was tested on saliva, it softened the material, while the immersion in gastric fluid diminished the resistance to mechanical force by the degradation of the polymer and double bonds rupture by the pH 1.2 of the fluid. Afterward, they tested the sensor applications by designing an inductor-capacitor structure composed of an interdigitated capacitor with five electrodes and a helically wound inductor. Then, gold was vapored to obtain the conductive structure at the top [20].

Organic and hydrogels, semisolid polymers based on three-dimensional networks, possess potential applications in organic electronics and photovoltaics: A hydrogel composed by poly[(R)-3-hydroxyundecanoate-co-(R)-3-hydroxy-10-undecenoate] (PHU10U—obtained from a *Pseudomonas* strain), with polyethylene glycol dithiol (PDT) as a photo-crosslinker [149] demonstrated good biocompatibility. In addition to the biomedical applications of organogels, such as drug and vaccine delivery matrices. They may also be used in environmental protection [150] and electronically active soft materials [149,151].

Carbon-based materials may be used as electrode materials in supercapacitors, defined as devices that store electrical energy in an electrical field, which are used in pieces of equipment that require fast charge/discharge cycles, such as vehicles and elevators, and energy storage, among others [152]. Common polymer precursors used usually collapse and aggregate during high-temperature treatment, resulting in low electrical conductivity and efficiency. Because of this, the carbon-based material may be improved post-heat treatment, but the compounds used to improve the pore structure may be corrosive, as well as only affecting the surfaces of the carbon [153]. A synergy was found between PHA and urea, which contributes to the incorporation of nitrogen within the carbon structure, producing a polymer with unique pore structures and higher porosity. This methodology is considered under the concept of in situ self-modification, in which urea modifies the PHA structure inside the bacterial strain, allowing the production of a biopolymer with high specific capacitance and energy density, excellent rate capability, and long cycling stability as supercapacitor electrode materials [152].

The self-modification strategy, which could be used in other areas of biotechnology, allows researchers to control PHA bacterial accumulation by regulating the composition of the cultivation media.

To improve the biopolymer’s properties, functional groups may be modified; for example, the functional groups of PHA (-OH (hydroxyl), -CO- (carbonyl), and -COO- (carboxyl)) generate gases, such as carbon mono- and dioxide and water, in the process called carbonization in high temperature. The carbonization process is used when complex carbonaceous substances are broken down into elemental carbon and chemical compounds (which also have carbon) by using heat. In this case, carbonization exfoliates the porous carbon, obtaining a larger surface area and highly porous structure, so when urea is added into the bacterial cell, it combines with PHA forming several nitrogen-containing functional groups, which release volatilized species during carbonization [154]. Specifically, the oxygen-containing functional groups interact with the nitrogen-based groups, thus incorporating nitrogen into the PHA matrix, producing a network with higher surface area and good conductivity, and as aimed, as an improvement in its performance as an electrochemical capacitor.

### 3.8. Other Uses

Other biocomposites, such as PHB/hydroxyapatite, were initially used as matrixes for bone implants and have been used for other biomedical purposes because of their good biocompatibility, osteoconductive and bioactivity [155], as well as high rigidity and low elasticity [156]. Chitin, found in shells of insects, lobsters, and shrimps, can interact with PHB to create a fully biodegradable composite [157] with potential applications in the agriculture and biomedical fields. Blends between PHB and natural fiber composites have better mechanical properties such as strength, stiffness, and others [79,158] and lowering production costs; for example, a blend between PHB/kenaf fiber and wood flour, enhancing tensile strength and elasticity [159], PHB and beech wood flour, obtaining a biopolymer with higher thermal stability, making them suitable for disposable articles [160], a film composite, based on PLA-PHB-limonene, with an adequate gas barrier, water resistance, and transparency for food packaging applications [161], among others. Also, biopolymers may be degraded in water, allowing their use in fresh fruit packaging or industrial products with short shelf life requirements [159,160,161].

#### 3.8.1. As Flame Retardant

An interesting function of PHA is as a flame retardant [162]. A composite based on poly(3-hydroxybutyrate-co-3-hydroxyvalerate)/poly(butylene adipate-co-terephthalate; PHBV/PBAT), enhanced its flame retardancy properties, accompanied by good thermal and mechanical properties [163]. Additionally, combinations with aluminum phosphinate (AlPi), nanometric iron oxide, and antimony oxide has been introduced. PBAT degrades by enzymatic degradation within weeks and is used to improve the elongation/toughness of bio-blends, as well as the reduction in tensile strength and modulus, compared to the unmodified polymer [164]. The addition of AlPi (as a flame inhibitor in the gas phase) and the metal oxide nanofiller (as a stabilizer of intermediate structures containing oxygen) helps in the flame-retardancy mechanism. To obtain a composite with relatively good mechanical properties and with fire retardancy properties, Gallo et al. (2013) proposed a bi-layer laminate system based on biodegradable PHA, both of them based on PHBV/PBAT: One of them with the addition of aluminum diethylphosphinate flame retardant and nanosized antimony oxide, while the second layer had the kenaf fibers addition [165].

#### 3.8.2. For Metabolic Pathways’ Analysis and Physiology

While not biomedical products per se, there is a need to understand how these polymers behave under physiological conditions in humans [166] and plants [167]. Little research has been made on the intracellular trafficking pathways of PHBHHx (3-hydroxybutyrate-co-3-hydroxyhexanoate) nanostructured materials. Biological studies can provide supportive information on the understanding of other spherical nanostructured materials as well. More recently, PHA as an electrospinning scaffold is also being investigated to produce sustainable in vitro models of biological barriers for investigating the physiopathological processes involved in the development of numerous diseases [166]. Sun et al. (2019) used Rab protein used as a marker to explore the intracellular trafficking mechanism of PHBHHx nanoparticles (NPs). When autophagy inhibitors and chemical drugs were packaged in PHBHHx and, used simultaneously, the volumes and weights of the tumors were significantly decreased. Additionally, NPs were internalized in cells mainly via clathrin endocytosis and caveolin endocytosis. Besides the classical pathways, they discovered two new pathways: the micropinocytosis early endosome (EEs)-micropinocytosis-lysosome pathway and the EEs-liposome-lysosome pathway. NPs were delivered to cells through endocytosis recycling vesicles and GLUT4 exocytosis vesicles. Similar to other nanoparticles, NPs also induced intracellular autophagy and were then degraded via endolysosomal pathways [166,167,168].

The overproduction of single-use plastic has generated an environmental crisis that has led us to look for compounds that degrade quickly, leaving no trace. In the environment, PHA polymer is degraded aerobically and anaerobically, using it as a carbon source. PHA depolymerases are produced by both bacteria and fungi. The degradation process depends on the characteristics of the PHA, such as crystallinity, size, thickness, molecular mass, and composition, but it also depends on temperature, pH, and humidity [169]. The time frame for biodegradation is critically important in defining the suitability of plastic for a particular end-of-life management technology or its likely fate in the environment. If the micro-organisms and, therefore, key enzymes involved in biodegradation are present, but the rate of degradation is so low that it does not differ significantly from non-degradable counterparts, then the limited biodegradability does not offer a benefit to the environment or the management of biodegradable waste. According to currently employed international standards, the proposed time frame for biodegradation in water environments is 56 days, and in soil, up to 2 years [170,171].

The key hydrolytic enzymes involved in the microbial degradation of PHA are depolymerases [170]. A variety of microorganisms have been identified that possess the enzymatic tools to degrade different types of PHAs, such as the genera *Achromobacter*, *Nocardia*, *Variovorax*, and *Streptomyces*. *Streptomyces* is the genus that produces the largest amount of PHA depolymerases. In combination with other polymers, it can degrade faster. For example, the use of tributyrin, dodecanol, lauric acid, and trilaurin increases degradation when used at 1% *w*/*v*. Much higher concentrations act as retardants.

Inside the body, the PHA polymer is broken down by the enzymatic action of proteins present in tissues and blood. Pancreatin, gastric juice, and bovine serum have been effective in degrading PHA microspheres. Pancreatin is composed of a mixture of esterases, lipases, amylases, and proteases. The active site of pancreatic lipase contains amino acid sequences, which are also present in the active site of PHB depolymerase [169].

It has been suggested that rapid PHBHHx degradation occurs in the amorphous region rather than in the crystalline region. At the same time, the in vivo hydrolysis of PHB was found to start from a random chain scission both in amorphous and crystalline regions of the polymer matrix. They also observed mild inflammatory effects of the implanted compound in the mice [172].

It is important that the degradation metabolites of the polymers, such as oligomers and monomers, are not harmful to surrounding tissue. PHB is mainly degraded under acidic conditions through acid hydrolysis of the ester bond in higher organisms. Oligomers produced by degradation have been shown to have positive effects on mouse fibroblast growth and are not detrimental to murine beta cells. The 3HB monomer is one of the three major ketone bodies that have regulatory effects on insulin secretion. 3HB has also been associated with high cellular calcium permeability by forming non-selective ion channels across the cell membrane. 3HB has a protective effect on dopaminergic neurodegeneration associated with increased mitochondrial respiration and ATP production [169].

#### 3.8.3. As Cleaning Material

Organogels may be used as cleaning materials, particularly when a solvent-controlled delivery system is present to avoid substrate damage. PHB-based gels with green solvents were proposed to be a new cleaning system [173] by producing a compound that combines gels with other mechanically resistant materials to create a composite. Jia et al. (2020) developed an organogel based on poly(3-hydroxybutyrate) and γ-valerolactone for restoration, aiming to improve the removal process of varnish from a painting. For these purposes, the composite is formed by a PHA moiety encased, the core that contains the active solvent, in two external layers of nonwovens made of submicrometric fibers, either poly(vinyl alcohol) or polyamide. The composite material showed excellent cleaning properties by completely removing the varnish layer [174].

#### 3.8.4. For Environmental Protection

By using PHA bionanoparticles, the immobilization of the lytic enzyme of a methanogen-integrated provirus, PeiR, inhibited an exceptionally broad range of different rumen methanogen strains in pure culture while significantly reducing methane production for several days, showing promising results that would decrease greenhouse gases [175].

#### 3.8.5. Other Uses

Additional PHA uses may be found in Table 2.

## 4. Conclusions: The Future of PHA

PHAs are a great opportunity; the chemical diversity of monomers and copolymers, as well as tunable physical properties, but also a challenge. Researchers aim to optimize biopolymer production to cheapen the costs. Further processing technologies, such as 3D printing, are a reality. Bioinformatics, such as machine learning, has been one of the best tools for enhancing PHA production, providing tools and strategies for property prediction and materials design via creating models built on reliable past data. Data may help in the prediction of structure and fingerprinting schemes based on the topology, shape, and charge/polarity of specific chemical units [194,195].

Also, molecular dynamics simulations may be done to predict chemical trends in the rheological and mechanical properties of biopolymers by analyzing Young’s modulus, yield stress, and deformation simulations in different types of PHA. All these predictions, for example, elucidated that both Young’s modulus and yield stress decrease with the increase in the number of carbon atoms in the side chain as well as in the polymer backbone, as well as the effect of functional groups within the mechanical properties of the polymer, which increases Young’s modulus and yield stress. Bioinformatics provides insights for rational design rules to design and modify mechanical properties and, in the future, adapt biopolymers to industry [195]. Machine learning has now entered the field; for example, Bejagam et al. used it to predict the melting temperatures of PHAs by combining curated data sets containing molecular weights and polydispersity indexes, together with descriptors on topology, shape, and charge/polarity of specific motifs [196], while other research groups also focused on glass transition temperature Tg values [194]. This type of research will add and optimize systems biology and evolutionary engineering processes to address polymer design with multiobjective optimization challenges.

The future is bright: Bioplastics global market size estimates have an increase projection from USD 9.2 billion to USD 20 billion by 2026, while the global bioplastics production capacity was over 2.1 million tonnes in 2020, which the projected expansion to 2.9 million tonnes by 2025 [124].

Genes involved in PHA production are well understood, so foreign proteins fused on the N- and/or C-terminus may allow attaching of one or more enzymes on the particle surface [197]. Although there are some unknown aspects of PHA particle formation, as stated by Wong and Rehm (2018), in which the protein fusion to phaC influences the PHA production yield, particle size distribution, and surface charges, the density of fused proteins on the PHA particles varied, among others [107].

Wong et al. offer a great summary of the advantages and limitations of PHAs: Polyhydroxyalkanoate technology allows for great production yields, monomer and polymer functionalization, stability, versatility, and ductility, as well as it may be obtained from different recombinant expression systems, presenting an enhanced shelf-life and biodegradability, but some physicochemical properties, such as particle size and distribution, are difficult to control [197,198,199].

The market price is still a problem, so requirements to develop novel (or enhances) producer strains are a must. They can be by systems biology, synthetic biology, protein engineering, and evolutionary engineering [139]. All the efforts should be focused on the improvements in the titer, yield, productivity, structure/theology of the biomaterial, and processability of natural and non-natural polyesters to increase the number of PHA applications in the industry. Plastic contamination is more of a concern now due to microplastics: Barron & Sparks (2020) describes that several research papers have evaluated PHA biodegradation in marine environments [198]. For example, Dilkes-Hoffman et al. (2019) revised marine degradation studies of PHA-based products, finding that PHA films with a thickness of 0.2% biodegraded within several months in marine environments [199].

PHA polymer is a promising compound in the development of several different lines. From nanotechnology for the production of films with nanometric fibers, food packaging that allows its preservation, with antimicrobial and breathable properties, to medical devices such as drug delivery or material to make masks. Much has been raised that biomaterials would be the definitive solution to the high demand for plastic products and clinical material (single use). This is because PHA is a biopolymer that is considered interesting since it can be degraded naturally. However, this point is only valuable if the product is arranged in the right place to degrade after use, it must be in contact with microorganisms that have PHA depolymerases. This is a challenge in itself.

On the other hand, it has also been suggested that because it is biocompatible, it can be widely used for critical surgical materials [200]; however, certain reports have revealed that this polymer can cause acute and chronic inflammation [15,201].

It remains to study these inflammatory processes and the various responses that PHA produces in vivo. For this reason, we propose that PHA be used more in industrial processes than in medical devices, where the product comes into direct contact with the interior of the patient or their critical fluids, such as blood. This is while continuing to study forms of PHA extraction that allow high purity levels and the copolymers generate the least possible effect in the body due to their degradation processes.

Summarizing, the present and future of PHA are promising. Despite its production costs still being higher than petroleum-based plastics, the versatility given by gene engineering and bacterial production allows for greater PHA yields and the ability to change and functionalize monomers and polymers to control the mechanical, thermal, and rheological properties according to the monomer, allow researchers to find novel uses and stimulate the research on this interesting family of polymers in industry.

## Figures and Tables

**Figure 1 molecules-27-08351-f001:**
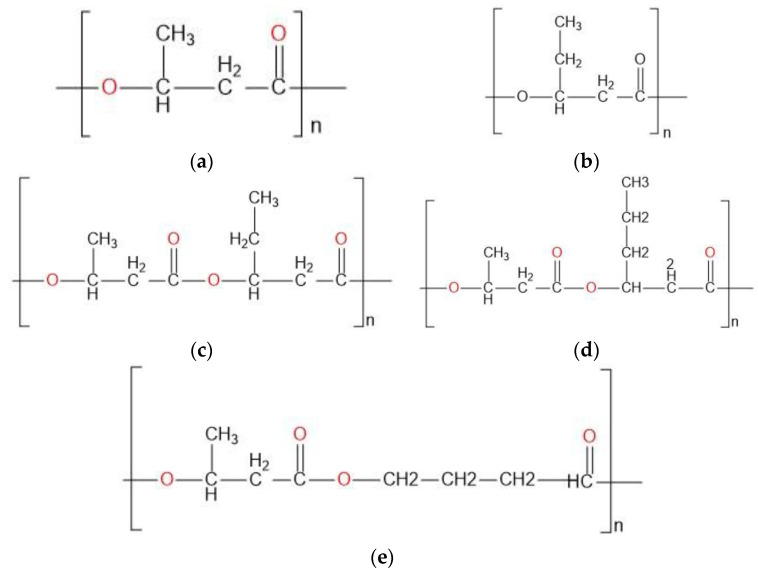
Polyhydroxyalkanoates used in industry. (**a**) Polyhydroxybutyrate (PHB); (**b**) Polyhydroxyvalerate (PHV); (**c**) Poly (3-hydroxybutyrate-co-3-hydroxyvalerate)(PHBV); (**d**) Poly (3-hydroxybutyrate-co-3-hydroxyhexanoate; P(3HB-co-3HHx)); (**e**) Poly (3-hydroxybutyrate-co-4-hydroxybutyrate) (P(3HB-co-4HB)).

**Figure 2 molecules-27-08351-f002:**
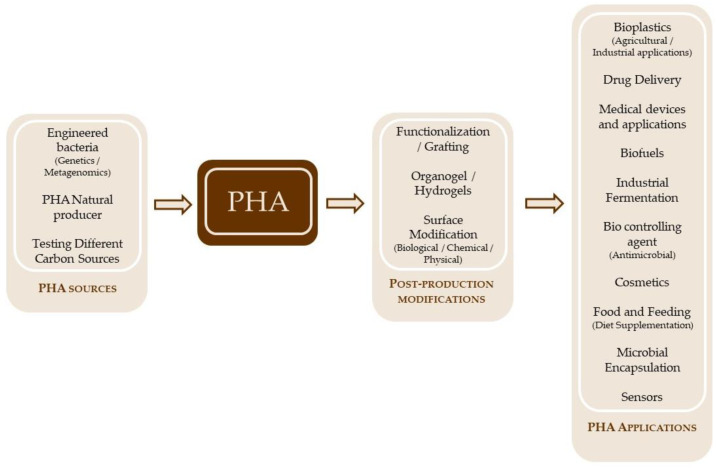
Generalities of PHA production.

**Table 1 molecules-27-08351-t001:** Main enzymes involved in PHB Synthesis. Information from [26].

Enzyme	Function	From
PhaA	3HB monomer-supplying enzymes. b-ketothiolase	*Ralstonia eutropha*
PhaB	3HB monomer-supplying enzymes. NADPH-dependent acetoacetyl-CoA reductase	*Ralstonia eutropha*
PhaC	PHA synthase	*Ralstonia eutropha*
PhaJ	(R)-specific enoyl-CoA hydratase. Catalyzes the generation of mcl-3HA-CoA from enoyl-CoA	*Aeromonas caviae*
PhaG	(R)-3-hydroxyacyl-acyl carrier protein (ACP)-CoA transferase	several pseudomonads
PhaP	PHA granule binding	

**Table 2 molecules-27-08351-t002:** Other novel uses for PHA biopolymers.

Biopolymer Blend	Aim/Proposed Use	Reference
mcl-PHA	Ibuprofen impregnation on mcl-PHA	[176]
mcl-PHA terpolyester	As biomaterial & ink	[177]
Fluorinated polyhydroxyalkanoates	As a biomaterial coating	[178]
Polyhydroxyalkanoates and Cellulose Nanocrystals	As high-oxygen-barrier multilayer films	[179]
Polyhydroxybutyrate accumulation on *Shewanella marisflavi* BBL25	To evaluate performance in electricity production	[180]
Oxygen Plasma Treated-Electrospun Polyhydroxyalkanoate Scaffold	For hydrophilicity improvement and cell adhesion enhancement	[181]
PHB nanofibers	For enzyme immobilization	[182]
Polyhydroxyalkanoates (PHAs; review)	For biofuel and biorefineries	[183]
Sulfonated Polyhydroxyalkanoate and Tannic Acid Derivative	As an antioxidant network	[184]
Tyrosinase-functionalized polyhydroxyalkanoate bio-beads	For bisphenol analog degradation	[185]
Polylactic acid/polyhydroxyalkanoates active film containing oregano essential oil	To enhance the quality and flavor of chilled pufferfish (*Takifugu obscurus*) fillets—as essential oil carriers	[186]
Polyhydroxyalkanoates-Based Nanoparticles	As essential oil carriers	[187]
Layered bacterial nanocellulose-PHBV composite	For food packaging	[188]
Poly(3-Hydroxybutyrate-co-3-Hydroxyvalerate) with Fruit Pulp Biowaste Derived Poly(3-Hydroxybutyrate-co-3-Hydroxyvalerate-co-3-Hydroxyhexanoate)	As organic, recycled food packaging	[189]
Cinnamaldehyde-Loaded Mesoporous Bioactive Glass Nanoparticles/PHBV-Based Microsphere	To prevent bacterial infection and promote bone tissue regeneration	[190]
Biocompatible Polyhydroxyalkanoate/Chitosan-Tungsten Disulphide Nanocomposite	For antibacterial and biological applications	[191]
Phage Lytic Enzymes Displayed on Tailored Bionanoparticles	To inhibit *Listeria monocytogenes* growth	[192]
Mycobacteriophage Endolysins Fused to Biodegradable Nanobeads	To mitigate mycobacterial growth in liquid and on surfaces	[193]

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
