# Peer review of "Novel Production Methods of Polyhydroxyalkanoates and Their Innovative Uses in Biomedicine and Industry"

_molecules, 2022, doi:10.3390/molecules27238351_

Round 1

Reviewer 1 Report

The review of Pamela Pavez and Guillermo Fernandez-Bunster is about polyhydroxyalkanoates and its application not only in medicine but for making new materials, such as biofilms, sensors etc. The authors refer to 134 sources, and the review itself consists of sixteen pages of plain text, one table. The review is based on a brief description of the work that has been done in this area.

The authors have done a great job. But in this form, I can not choose minor or even major revision.  

https://www.mdpi.com/about/article_types

“Reviews offer a comprehensive analysis of the existing literature within a field of study, identifying current gaps or problems. They should be critical and constructive and provide recommendations for future research.”

In the current manuscript there is only a description of the published papers. The task of the authors of the review is to highlight the problem, or show young researchers where to start. Based on the review, we can conclude that PHA is studied by many groups around the World. In its current form, it is inconvenient to use the review. To find out information about PHA, it will be more convenient to enter a query “PHA uses” into Google Scholar, sort by date and read abstracts of published articles.

In order to allow this review to be published, it is necessary to analyze sources, make a summary table or diagram that would show the trends. As for example done in existing reviews about PHA:

https://doi.org/10.1111/pbi.12039

https://doi.org/10.1039/B812677C

https://doi.org/10.1039/B209687K

It is worth adding schematically what is PHA, PHB, PHV.

Despite the fact that the review is called “Polyhydroxyalkanoates: Uses beyond medicine”, most of it is devoted specifically to medical use, like

3.3. Drug Delivery of Natural Compounds

3.8.2. For metabolic pathways’ analysis and physiology

3.8.5. Other uses, table 1

Ibuprofen impregnation on mcl-PHA; Cell Adhesion Enhancement; As Antioxidant Network; To prevent bacterial infection and promote bone tissue regeneration

Reviewer 2 Report

In this manuscript review G. Fernandez-Bunster and P. Pavez present a summary of the potential use of PHAs beyond the most common uses in medicine as well as in biosensors, cosmetics or drug delivery etc. It is a readable and lucid article that has the potential to engage a wider audience. 

I would recommend supplementing their research with the latest outputs of evolutionary engineering studies with respect to PHAs production. In the final summary, I lack more striking conclusions and consideration stemming from the challenge of PHAs biodegradation. The manuscript is of general interest and I, therefore, recommend to accept the article after minor revision.

Round 2

Reviewer 1 Report

Dear authors,

You have indeed greatly remastered your manuscript, the added graphical information and tables make the review more convenient and clearer. I am pleased with the work you have done and I hope you understand my first reject decision.

I wish your team new scientific results!

Best wishes